# Global Social Sustainability and Inclusion: The "Voice" of Social and Environmental Imbalances

**Andriy Krysovatyy [1], Iryna Zvarych [2],\***  **, Oksana Brodovska [3] and Roman Zvarych [4]**

1   West Ukrainian National University, 46009 Ternopil, Ukraine
2   International Economics Department, West Ukrainian National University, 46009 Ternopil, Ukraine
3   Limited Liability Company Financial Company MAGNAT, West Ukrainian National University, 46009 Ternopil, Ukraine
4   International Economic Relations Department, West Ukrainian National University, 46009 Ternopil, Ukraine
\*   Correspondence: irazvarych@gmail.com; Tel.: +380-967096889

**Abstract:** Background: Global environmental and social research strengthens the protection of people and the environment, develops national capacity for social and environmental management and enables significant progress in terms of transparency, accountability, nondiscrimination and public participation. The support of the general public plays a key role, as it contributes to making public institutions more transparent, accountable and efficient and promotes ground-breaking solutions to complex development challenges. Citizen engagement seems particularly vital throughout the crises such as the COVID-19 pandemic, as the efficiency of response efforts may frequently depend upon micro-level behavioral changes. The objective of this paper is to provide a complex evaluation and rating of countries based on the social component of the global inclusive circular economy, taking into account the shocks and reverberations experienced by the economy as a whole caused by the COVID-19 and war in Ukraine. The results are presented as a global ranking of countries based on the social component of the global inclusive circular economy. They confirm the high value of this component in the integrated indicator, which validates the hypothesis that inclusiveness is a necessary aspect of the global circular economy. The research results identify the countries capable of offering the best management solutions to social disbalances and other weaknesses, as well as the countries in need of model examples to tackle these issues.

**Keywords:** social inclusion; global sustainability; global inclusive circular economy; poverty

## 1. Introduction

From a broad socio-philosophical perspective, inclusion can be defined as a special form of being wherein people with disabilities (disabled people) lead lives similar to those of average individuals, for or against which society and its subsystems (including educational institutions) advocate, and for which these and other members of society have the right of free choice.

Global inclusive circular economy as a concept designates a significant place to "inequality", since its increase is a considerable threat to sustainable economic development. However, the relevance of the concept of "justice" for the economy is quite controversial.

In every country, some sections of the population face obstacles that hamper them from being able to actively and equally take part in social, political and economic activities. Such people face exclusion due to biased legal systems, land and labor markets, as well as prejudiced or stigmatizing attitudes, viewpoints and misconceptions that result in discrimination. As a rule, discrimination is often centered around one's sex, age, place of residence, occupation, race, ethnic origin, religion, citizenship status, disability, sexual orientation, gender identity (SOGI) and the like. Such social exclusion deprives people of dignity, safety and the opportunity to live a full and prosperous life. If the key causes of these structural exclusions and discriminatory behavior are not addressed, sustaining

the desired inclusive growth and rapid poverty reduction will be difficult (Daly 2001; Commoner 1992).

The methodology for measuring inclusive growth according to these types of structures necessitates developing the idea of how the social welfare scheme functions, which makes it possible to measure the distribution of opportunities among the population, while paying special attention to educational and medical needs (Krysovatyy et al. 2018b, 2018c). In general, the definition of inclusive growth inherent in process-oriented frameworks is broader than those that focus on end results. According to the researchers, the emphasis on the participation and contribution of all groups takes different forms. For example, additionally consider investment opportunities, and mostly focuses on education, health, nutrition and social integration, since the COVID-19 pandemic has drawn attention to these deep-seated systemic inequalities. As the aftermath of COVID-19 still lingers globally, it is important to understand the differential and amplified effect of the pandemic on the most disadvantaged and discriminated population groups, including women, disabled people, unemployed young people, sexual and gender minorities, elderly people, indigenous peoples, as well as ethnic and racial minorities. To illustrate, a great number of people with disabilities have health conditions which make them especially susceptible to severe COVID-19 symptoms. Women and children have reported rising levels of domestic violence due to the quarantine and amplified tension at home (Matviychuk-Soskina et al. 2019). In many cases, there is historical evidence of people experiencing obstacles when trying to access healthcare systems due to their ethnic or racial identity. Such groups have demonstrated higher mortality rates than the average, as they have difficulties gaining access to information about the pandemic, as well as access to equitable care and vaccines. The problem of global social inclusion and inclusive economy arose during the study of recycling and circular economy in general. After all, people are part of nature (Daly 2001) and economic problems are inextricably linked with ecology, ethics and social approaches. The problem of harmonizing life on earth in complete peace is described in the modern concept of inclusiveness (Commoner 1992). The problem of social sustainability and inclusion was introduced for the first time in the early work of Commoner (1992) "Make peace with the Earth". Daly (2001) continued to develop this idea. Carnemolla et al. (2021) presented the article "Towards inclusive cities and social sustainability: A scoping review of initiatives to support the inclusion of people with intellectual disability in civic and social activities", which strengthened this idea of studying the influence of the social component in general human rights. Malik et al.'s (2022) article "Role of social sustainability for financial inclusion and stability among Asian countries" insisted that problems of this nature are highly prevalent in Asian and African countries. So, the classification or regional view on the problem of social sustainability and inclusion will be the underlying idea for future research. Analysis of the reviewed works brings to mind considerations of global permanent inclusion and the role of different countries in this process. What factors, in one way or another, affect the isolation of the social component in global sustainable inclusiveness? Which countries have the most pronounced social component? Is there a rating of such countries? The authors contemplated such questions and offered answers in this article.

The novelty of this article lies in its focus on the ranking of countries according to the social component, based on the authors' own methodology, and the emphasis on social inclusiveness.

For the first time, this paper presents objective evidence in the form of a rating of countries based on the social component and loosely correlated derived indicators. Thus, the latter do not make a significant impact on solving disbalances in the studied countries. Such analysis is conducted for the first time and is therefore useful to the top managers and higher-up officials who shape state policy.

This topic is important now, and it will undoubtedly be important in the near future in light of the ongoing COVID-19 pandemic, the war in Ukraine and mass migration of Ukrainians to other countries. All of these factors exert social pressure on the local populace, exacerbating gender inequality and violence. This topic is an inexhaustible well

of questions and answers. Thus, we must set careful limits to our research, as we have chosen to focus on the social component, i.e., social inclusiveness, only.

## 2. Materials and Methods

If left unchecked, exclusion of vulnerable population groups may prove fairly costly. At the microeconomic level, the outcomes that are analyzed most frequently include wage loss, lifetime earnings, lower educational attainment and other issues in workplaces. At the macroeconomic level, the economic cost of social exclusion is reflected in lost gross domestic product (GDP) and human capital wealth (World Bank 2022). On a global scale, experts estimate the loss of human capital because of gender inequality alone at USD 160.2 trillion. Observed levels of poverty are still higher for African countries than any other region; for example, this indicator is 2.5 times lower for Latin America. Further, in developing countries, 90% of children with disabilities do not have access to even basic education. Combating exclusion, violence and discrimination in this regard is particularly difficult (World Bank 2022).

Alienation is dangerous in that, over time, it aggravates social tensions and increases the incidence of fighting, cruelty and hostility. This, in turn, causes long-term social and economic losses.

Social inclusion is vital to achieving two goals: tackling extreme poverty and increasing public welfare (Figure 1). The funding and investments for all innovative projects are managed within a social and environmental framework that highlights the critical importance of social inclusion when it comes to development interventions by international institutions and, therefore, the achievement of sustainable development.

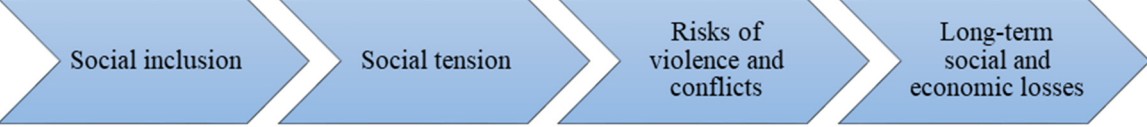

**Figure 1.** Key features of social sustainability and social inclusion.

That is why the flagship of international organizations, the World Bank, prioritized comprehensive recovery after the COVID-19 pandemic. In addition, the recent final report on IDA20 restocking increased the focus on inclusion. Specifically, 14 out of 41 policy commitments clearly refer to inclusion. Indeed, the problems of today should be viewed as opportunities for tomorrow, impetus to redesigning systems toward more inclusive solutions that will create a more resilient society prepared for potential crises of any kind, whether in healthcare or environmental management, caused by natural disasters or human dissatisfaction.

An emerging strategic goal of social sustainability and inclusion involves ways of addressing systemic and deep-rooted inequalities and providing all people with access to new and existing opportunities.

Social stability and inclusivity foster inclusive and sustainable societies in which responsive governments hear and value public opinion. This contributes to the short-term and long-term capacity for maintaining sustainable growth and reducing poverty. The starting point here is the citizens themselves, their values and goals, their communities and workplaces. Inclusive economic development policies are all too often aimed at national and regional governments or the private sector, rather than community development of marginalized or vulnerable populations, of those affected by conflict or those living in remote areas. Through a whole-of-society approach, we focus on both strengthening communities and improving their interactions with the private sector and the government. To this end, work should be carried out through producer associations, private actors and, where possible, institutionalized citizen involvement. The idea is to establish dynamic communities in which all residents can reach their full potential, both in terms of professional fulfillment and personal growth, thanks to a favorable business climate (World Bank 2022).

### 3. Results

The input data are composed of indicators from 1995 to 2020, taken from the statistical databases of the Organization for Economic Cooperation and Development and the relevant 28 member countries. The analysis was conducted using the GNU Regression, Econometrics and Time-series Library (Library for regressions, econometrics and time series)—an applied software package for econometric modeling, which is part of the GNU project.

Table 1 shows the ranking of countries in the research sample based on calculations centered around the social component of the global inclusive circular economy (Krysovatyy et al. 2018a; Matviychuk-Soskina et al. 2019).

$$\text{Social component}: \vec{x}_2 = \{x_{2i}\}, \ i = 1, \ldots, 5$$

where

$x_{21}$—welfare costs of premature death from lead exposure, equivalent to GDP;

$x_{22}$—population with access to improved sanitation (percentage of total population);

$x_{23}$—population with access to purified spring drinking water (percentage of the total population);

$x_{24}$—population connected to water supply networks (percentage of the total population);

$x_{25}$—population connected to water supply networks with preventive disinfection (percentage of the total population).

Analysis of the calculations has revealed the weakly correlated indicators for the economic component in different countries:

- Population with access to improved sanitation (percentage of the total population): *Denmark, South Korea;*
- Welfare costs from premature death from lead exposure, equivalent to GDP: *Luxembourg, Germany, Japan.*
- Population with access to purified spring drinking water (percentage of the total population): *Great Britain, Canada, Poland, Czech Republic.*
- Population connected to water supply networks (percentage of the total population): *Estonia, Israel, Mexico, Netherlands.*
- The population connected to water supply networks with preventive disinfection (percentage of the total population): *Turkey, South Africa, China.*

**Table 1.** Analysis results for the social component of the GICE indicator.

| Country | Regression Function | % Comp. | Weakly Correlated Indicators |
|---|---|---|---|
| 1 | 2 | 3 | 4 |
| Australia | $y(\vec{x}_2) = 0.385{\cdot}x_{21} + 0.469{\cdot}x_{22} + 0.469{\cdot}x_{23} - 0.45{\cdot}x_{24} + 0.456{\cdot}x_{25}$ | | 87% |
| Austria | $y(\vec{x}_2) = -0.445{\cdot}x_{21} + 0.460{\cdot}x_{22} + 0.466{\cdot}x_{23} + 0.456{\cdot}x_{24} + 0.406{\cdot}x_{25}$ | | 90% |
| Belgium | $y(\vec{x}_2) = -0.442{\cdot}x_{21} + 0.448{\cdot}x_{22} - 0.449{\cdot}x_{23} + 0.448{\cdot}x_{24} + 0.449{\cdot}x_{25}$ | | 89% |
| Greece | $y(\vec{x}_2) = 0.39{\cdot}x_{21} + 0.468{\cdot}x_{22} - 0.458{\cdot}x_{23} + 0.451{\cdot}x_{24} + 0.464{\cdot}x_{25}$ | | 90% |
| Denmark | $y(\vec{x}_2) = -0.449{\cdot}x_{21} + 0.520{\cdot}x_{23} + 0.498{\cdot}x_{24} + 0.529{\cdot}x_{25}$ | $x_{22}$ | 87% |
| Great Britain | $y(\vec{x}_2) = -0.574{\cdot}x_{21} + 0.588{\cdot}x_{22} + 0.570{\cdot}x_{25},$ | $x_{23}$ | 95% |
| Estonia | $y(\vec{x}_2) = -0.493{\cdot}x_{21} + 0.506{\cdot}x_{22} - 0.506{\cdot}x_{23} + 0.496{\cdot}x_{25}$ | $x_{24}$ | 97% |
| Israel | $y(\vec{x}_2) = -0.497{\cdot}x_{21} + 0.504{\cdot}x_{22} + 0.494{\cdot}x_{24} + 0.505{\cdot}x_{25}$ | $x_{24}$ | 98% |

**Table 1.** *Cont.*

| Country | Regression Function | % Comp. | Weakly Correlated Indicators |
|---|---|---|---|
| 1 | 2 | 3 | 4 |
| Canada | $y(\vec{x}_2) = 0.481 \cdot x_{21} + 0.52 \cdot x_{22} + 0.457 \cdot x_{23} + 0.538 \cdot x_{24}$ | $x_{23}$ | 83% |
| China | $y(\vec{x}_2) = 0.446 \cdot x_{21} + 0.457 \cdot x_{22} + 0.43 \cdot x_{23} + 0.456 \cdot x_{24} + 0.445 \cdot x_{25}$ | $x_{25}$ | 92% |
| Latvia | $y(\vec{x}_2) = -0.448 \cdot x_{21} + 0.456 \cdot x_{22} + 0.454 \cdot x_{23} + 0.437 \cdot x_{24} + 0.441 \cdot x_{25}$ | | 95% |
| Lithuania | $y(\vec{x}_2) = 0.501 \cdot x_{22} + 0.501 \cdot x_{23} + 0.499 \cdot x_{24} + 0.5 \cdot x_{25}$ | | 99% |
| Luxembourg | $y(\vec{x}_2) = -0.493 \cdot x_{21} + 0.509 \cdot x_{22} + 0.509 \cdot x_{23} + 0.49 \cdot x_{25}$ | $x_{21}$ | 96% |
| Mexico | $y(\vec{x}_2) = -0.503 \cdot x_{21} - 0.503 \cdot x_{22} - 0.504 \cdot x_{23} + 0.49 \cdot x_{25}$ | $x_{24}$ | 97% |
| Netherlands | $y(\vec{x}_2) = -0.51 \cdot x_{22} + 0.508 \cdot x_{23} - 0.462 \cdot x_{24} + 0.518 \cdot x_{25}$ | $x_{24}$ | 90% |
| Germany | $y(\vec{x}_2) = 0.44 \cdot x_{21} - 0.518 \cdot x_{22} - 0.494 \cdot x_{23} + 0.292 \cdot x_{24} + 0.457 \cdot x_{25}$ | $x_{21}$ | 83% |
| New Zealand | $y(\vec{x}_2) = 0.444 \cdot x_{21} + 0.553 \cdot x_{22} + 0.451 \cdot x_{23} + 0.542 \cdot x_{24}$ | | 75% |
| South Africa | $y(\vec{x}_2) = -0.707 \cdot x_{21} + 0.707 \cdot x_{24}$ | $x_{25}$ | 89% |
| South Korea | $y(\vec{x}_2) = 0.026 \cdot x_{21} + 0.498 \cdot x_{22} + 0.503 \cdot x_{23} + 0.502 \cdot x_{24} + 0.495 \cdot x_{25}$ | $x_{22}$ | 79% |
| Poland | $y(\vec{x}_2) = -0.429 \cdot x_{21} + 0,451 \cdot x_{22} - 0.452 \cdot x_{23} + 0.451 \cdot x_{24} + 0.452 \cdot x_{25}$ | $x_{23}$ | 97% |
| Turkey | $y(\vec{x}_2) = 0.563 \cdot x_{21} - 0.624 \cdot x_{22} - 0.536 \cdot x_{24} + 0.078 \cdot x_{25}$ | $x_{25}$ | 76% |
| Hungary | $y(\vec{x}_2) = 0.501 \cdot x_{22} + 0.482 \cdot x_{23} + 0.511 \cdot x_{24} + 0.506 \cdot x_{25}$ | | 93% |
| Finland | $y(\vec{x}_2) = 0.427 \cdot x_{21} - 0.638 \cdot x_{22} + 0.641 \cdot x_{23}$ | | 76% |
| France | $y(\vec{x}_2) = -0.395 \cdot x_{21} + 0.464 \cdot x_{22} - 0.464 \cdot x_{23} + 0.460 \cdot x_{24} + 0.449 \cdot x_{25}$ | | 92% |
| Czech Republic | $y(\vec{x}_2) = -0.451 \cdot x_{21} - 0.454 \cdot x_{22} + 0.455 \cdot x_{23} + 0.447 \cdot x_{24} + 0.429 \cdot x_{25}$ | $x_{23}$ | 95% |
| Japan | $y(\vec{x}_2) = 0.214 \cdot x_{21} + 0.346 \cdot x_{22} + 0.525 \cdot x_{23} + 0.531 \cdot x_{24} + 0.526 \cdot x_{25}$ | $x_{21}$ | 85% |

Illegal and informal recycling activities, particularly in developing countries, with the practice of informal e-waste recycling in the disassembly process, frequently produce toxic emissions and spent acid, lacking correct treatment and oversight. It is this that results in problems with health and the environment, which highlights the relevance of the inclusive and social component to the methodology proposed for defining a global inclusive circular economy (Reznikova et al. 2019).

Trade flows of waste and scrap do not, in themselves, indicate an increase or decrease in pressure on the environment. The bottom line is whether such trade in waste and scrap is treated and recovered in an ecological environment and is consistent with circular economy goals (Wang et al. 2019).

As global value chains emerge and the formation of global circular added-value chains intensifies, eco-design and eco-labeling serve as some of the most fundamental steps in facilitating the transition to a global circular economy and breaking barriers to it.

The real value of the global waste market lies in the opportunity to create circular startups which integrate into global circular chains of added value.

The circular gap (gap circularity), which stems from the scarcity of natural resources, global population growth and overconsumption, may potentially be managed by reducing waste and rationally using resources (Zhou et al. 2014).

Based on the conducted calculations and identified indicators, we can determine the indicators that cause a decrease in the social component (Table 2) and the rank of the countries by the value of the social component of the global inclusive circular economy (Table 3). This allows us to conclude that this component holds significant weight in the integral index. Here, Belgium is the undisputed leader, followed by the Czech Republic, the USA, China, France, Greece, Austria and Australia. The rating is closed out by South Africa, with zero on social indicators. Notably, Ukraine is 25th among the 28 studied countries.

**Table 2.** Indicators that cause a decrease in the social component.

| Countries | Indicators |
|---|---|
| Australia | Population connected to water supply networks with preventive disinfection (percentage of the total population) |
| Austria | Welfare costs of premature death from lead exposure (equivalent to GDP) |
| Belgium | Population with access to purified spring drinking water (percentage of the total population) |
| Greece | Population with access to purified spring drinking water (percentage of the total population) |
| Israel | Welfare costs of premature death from lead exposure (equivalent to GDP) |
| Latvia | Welfare costs of premature death from lead exposure (equivalent to GDP) |
| Luxembourg | Welfare costs of premature death from lead exposure (equivalent to GDP) |
| | Welfare costs of premature death from lead exposure (equivalent to GDP) |
| Mexico | Population with access to improved sanitation, (percentage of the total population) |
| | Population with access to purified spring drinking water (percentage of the total population) |
| Germany | Population with access to improved sanitation, (percentage of the total population) |
| | Population with access to improved sources of drinking water, (percentage of the total population) |
| South Africa | Welfare costs of premature death from lead exposure (equivalent to GDP) |
| Poland | Welfare costs of premature death from lead exposure (equivalent to GDP) |
| | Population with access to improved sources of drinking water (percentage of the total population) |
| Turkey | Population with access to improved sanitation, (percentage of the total population) |
| Finland | Population with access to improved sanitation, (percentage of the total population) |
| France | Welfare costs of premature death from lead exposure (equivalent to GDP) |
| | Population with access to improved sources of drinking water, (percentage of the total population) |
| Czech Republic | Welfare costs of premature death from lead exposure (equivalent to GDP) |
| | Population with access to improved sanitation, (percentage of the total population) |

**Table 3.** Rating of countries according to the social component of the global inclusive circular economy.

| Attitude | Country | Indicator, $\lambda_{y(\vec{x}_1)}$ | Normalized Index |
|---|---|---|---|
| 1 | Belgium | 4.8185 | **1** |
| 2 | Czech Republic | 4.76 | **0.9807** |
| 3 | USA | 4.7321 | **0.9714** |
| 4 | China | 4.6212 | **0.9348** |
| 5 | France | 4.6083 | **0.9305** |
| 6 | Greece | 4.5131 | **0.8991** |
| 7 | Austria | 4.4996 | **0.8946** |
| 8 | Australia | 4.38 | **0.8551** |
| 9 | Lithuania | 3.9858 | 0.7248 |
| 10 | South Korea | 3.9389 | 0.7093 |
| 11 | Israel | 3.9173 | 0.7022 |
| 12 | Mexico | 3.8922 | 0.6939 |
| 13 | Estonia | 3.8876 | 0.6924 |
| 14 | United Kingdom | 3.8749 | 0.6882 |
| 15 | Poland | 3.8676 | 0.6857 |
| 16 | Luxembourg | 3.8424 | 0.6774 |
| 17 | Latvia | 3.7489 | 0.6465 |

**Table 3.** *Cont.*

| Attitude | Country | Indicator, $\lambda_{y(\vec{x}_1)}$ | Normalized Index |
|---|---|---|---|
| 18 | Hungary | 3.7426 | 0.6444 |
| 19 | Netherlands | 3.6259 | 0.6059 |
| 20 | Denmark | 3.5104 | 0.5677 |
| 21 | Japan | 3.4447 | 0.5460 |
| 22 | Canada | 3.319 | 0.5044 |
| 23 | Germany | 3.2441 | 0.4797 |
| 24 | Finland | 3.1894 | 0.4616 |
| **25** | Ukraine | **3.137** | **0.4443** |
| 26 | New Zealand | 3.002 | 0.3997 |
| 27 | Turkey | 2.2761 | 0.1598 |
| 28 | South Africa | 1.7926 | 0 |

The decentralized, informal economy makes up a significant portion of the global economy, employing an estimated 61% of all workers. In fact, including agriculture, the informal sector accounts for more than 90% of total employment in a number of African and Asian countries. Despite the low cost of labor in the informal sector, such employers must face highly efficient sorting and reverse logistics techniques at the "last mile" or the last point of the value chain. The workers deal with poor working conditions, insufficient income and, as a result, problems related to health. However, this decentralized, distributed nature of the informal economy modifies the economic system into a very organic and flexible organism. Therefore, social justice is driven by the need to improve the qualifications and organization of informal workers, to ensure that their working conditions improve. Education is one of the tools that can help to implement and prolong the concept of circularity and inclusiveness. It is an absolutely essential element for young entrepreneurs, designers and engineers, especially in terms of the technical characteristics of circularity and employment (i.e., job hunting) in the informal sector. Many communities are recycling waste into new products; in this way, we are talking about filling newly created gaps in the labor market, creating and providing circular workplaces of the future.

The results of the global ranking of countries based on the social component of the global inclusive circular economy have confirmed the high value of this component in the integrated indicator, which validates the hypothesis that inclusiveness is a necessary aspect of the global circular economy. Thus, Belgium is the undisputed leader, followed by the Czech Republic, the USA, China, France, Greece, Austria and Australia. South Africa closes the list with zero on social indicators. Notably, Ukraine is 25th among the 28 studied countries.

Economic and social background is not the only characteristic that the concept of inclusion is aimed at (Figure 2). Racial and ethnic minorities often face discrimination and exclusion. In Latin America, dark-skinned population is 2.5 times more likely to become poor. In addition, it is believed that minorities, such as those of African descent, face dire consequences more often whenever threats to public safety and welfare, such as the COVID-19 pandemic, emerge. Social inclusion and integration aim to ensure that every category of people, regardless of race or ethnicity, is heard, helped and supported, and thus can live full lives. This prompts real-time course correction so that solutions may be found in rapidly developing situations during or post crisis (Tables 4 and 5).

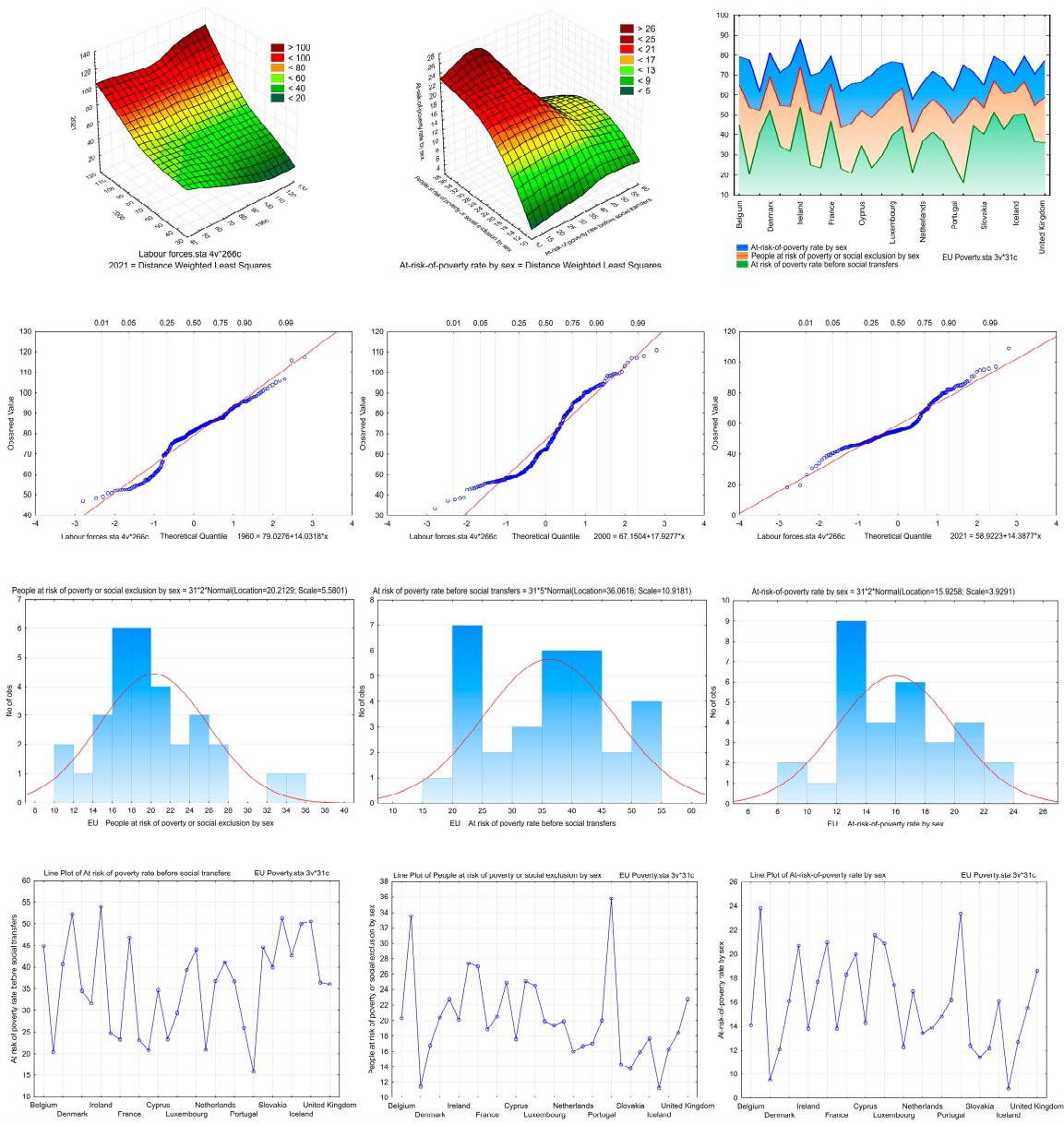

**Figure 2.** Visualized results of regression analysis of the social component.

**Table 4.** Activities of the World Bank towards Social Inclusion and Sustainability.

| № | Year of Implementation | Title |
|---|---|---|
| 1. | 2018 | The first Disability Inclusion and Accountability Framework; announced the World Bank's Ten Commitments to Accelerate Global Action for Disability Inclusive Development |
| 2. | | Multidisciplinary Task Force on Sexual Orientation and Gender Identity (SOGI) 20. (World Bank 2022) |
| 3. | 2016–2022 | The World Bank's Gender Strategy is about helping countries and companies achieve gender equality as a path to lasting poverty reduction and shared prosperity and security. Social Inclusion Platforms and Gender Platforms were created by the World Bank as a consolidated approach to social inclusion. Platforms in Afghanistan, India, Pakistan, Kenya, Somalia, Uganda and Ethiopia, among others, provide strategic support to social inclusion and gender project teams by sponsoring advanced technical and analytical work, facilitating innovation and knowledge sharing, and integrating activities into the Bank's operations (World Bank 2022). |

**Table 5.** Social-inclusion projects of the World Bank.

| 1. | Country | Response Program | Essence |
|---|---|---|---|
| 2. | Republic of Peru | Emergency Response for Venezuelan Migrants and Refugees. | Providing emergency economic support to help smooth the consumption of some of the most vulnerable Venezuelan migrants and refugees affected by the COVID-19 pandemic in selected cities in Peru. |
| 3. | Republic of Bangladesh | Crisis Response Project | Strengthening Bangladeshi government systems to improve access to essential services and increase social resilience of the population of Cox's Bazar district to disasters. |
| 4. | Democratic Republic of the Congo | Gender-Based Violence Prevention and Response Project | Reaching 795,000 direct beneficiaries over four years and providing assistance to victims of gender-based violence; changing social norms by promoting gender equality and behavior change through close partnership with civil society organizations. |
| 5. | India | Nai Manzil: Education and Skills Training for Minorities | Improving postsecondary completion and marketable skills training for targeted minority youth. |
| 6. | Nepal | Nepal's Integrated Platform to Prevent and Respond to Gender-Based Violence | Support capacity-building activities that support victims of GBV, including indigenous groups and other vulnerable communities. |
| 7. | Federal Republic of Nigeria | Nigeria for Women | Direct impact on 324,000 women beneficiaries through investments in comprehensive skills training, leveraging financial and technical resources and supporting policy dialogue on women's economic empowerment. It also aims to create and strengthen women's affinity groups to facilitate social networking and increase women's voice and participation in the economy. |
| 8. | Republic of Panama | Integral Development of Indigenous Peoples | Improving public services in indigenous territories based on indigenous peoples' own views and development priorities and strengthening capacity for governance and coordination between government and indigenous authorities, while respecting indigenous cultural identity and ways of life. |
| 9. | Papua New Guinea | Urban Youth Employment project | Providing urban youth with income from temporary employment opportunities and increasing their employment opportunities. |
| 10. | Republic of the Philippines | National CDD project | Empowering communities in target municipalities, ensuring their participation in local governance and helping them develop their capacity to design, implement and manage poverty reduction activities. |
| 11. | Egypt | Cairo Airport Terminal 2 Reconstruction Project | A review of airport design and costs to improve accessibility measures, making the new airport accessible to people with disabilities. |
| 12. | Republic of Honduras | Safer Municipalities of Honduras | Prevention of interpersonal violence. |
| 13. | Laos | The second phase of the Poverty Reduction Fund | Improving access to infrastructure, sanitation and nutrition. |
| 14. | Myanmar | National CDD program | Reached over seven million beneficiaries with over 18,000 sub-projects completed in 61 villages. Communities have built or renovated more than 3300 schools, built more than 8000 km of footpaths and access roads, and generated more than 6.4 million paid person days. |
| 15. | Nicaragua | Land Resources Management project | Reduction of registration time and transaction costs. |
| 16. | Federal Republic of Nigeria | Community and Social Development project | An evaluation of the impact of the CSDP showed a reduction in maternal and child mortality; increasing school enrollment and attendance; reducing the distance, cost and time of access to water, medical services and electricity; and increasing income from agriculture. The World Bank has approved an additional USD 75 million loan to expand social security and improve services for communities, especially internally displaced people, affected by the conflict in northeast Nigeria. |

**Table 5.** *Cont.*

| 1. | Country | Response Program | Essence |
|---|---|---|---|
| 17. | Republic of Peru | Mechanism of Allocated Grants | Maintained fieldwork and administrative process for 88 communities to process formal claims to the government regarding their ancestral land titles, achieving formal land titles for 14 communities as of March 2019. In addition, the project benefited 56 local communities through technical and financial support for 40 forestry sub-projects, 10 of which are managed by women. |
| 18. | Somalia | Project of the Special Financial Mechanism for Local Development | Strengthening government systems, visibility and legitimacy through the provision of basic infrastructure and services. In particular, this project supports the Ministry of Finance in the procurement and supervision of small capital grants identified by communities and newly created federal lands, and to strengthen the new federal architecture in a country emerging from 20 years of conflict. |
| 19. | Vietnam | Rural Sanitation and Water Supply Expansion Program | Ensuring measurable and equitable rural sanitation and water-supply benefits for ethnic minorities in 21 provinces. |
| 20. | Central African Republic | Project LONDO | Provides temporary employment for vulnerable populations and facilitates access to basic services in the CAR. This project will help triple the number of beneficiaries of labor-intensive public works, providing about 5 million paid working days in the country. The project also directly supports the implementation of the peace agreement signed in 2019. |

Social sustainability and inclusion—previously known as social development—are a testament to the current focus on overcoming persistent barriers to development, increasing attention towards groups with less or no access to economic and social opportunities, and promoting investments in inclusive growth.

Social sustainability builds inclusive and resilient communities where people are entitled to having a voice and successfully interacting with those in authority. This means that all people are empowered now and in the future. The combination of social, economic and environmental sustainability is critical for tackling poverty and ensuring welfare in these regions.

Social sustainability and inclusion measures predominantly aim to help people—without discrimination against any particular group—break the barriers hampering them from leading active and fulfilling lives, supporting their efforts to build a better future for themselves. This is achieved by cooperating with authorities, local communities, civil society, the private sector and other stakeholders to create more inclusive societies, empower citizens and build more sustainable and nonviolent communities.

The theoretical substantiation of these concepts is quite thorough and far-reaching; however, critical analysis makes it possible to distinguish two interrelated approaches:

- Involvement of the poor population as a partner in ensuring growth and achieving well-being ("exit" from poverty);
- Inclusive growth.

Inclusive growth makes it possible to involve most of the labor resources in effective economic activities and provide a higher standard of living for the majority of the population. Moreover, antidiscrimination measures and activities can ensure that people can benefit from economic growth through redistributive policies as passive participants, without actively contributing to increasing income or GDP. This is a common practice in most countries, poor and developing alike. In truth, the difference between these two perspectives (people as active or passive participants, as producers and consumers, as actors or clients) is not as obvious as it seems, since inclusive development involves people taking an active part in the process of political, social and economic change. However, the

growth forecasts for the poor population indicate that inclusiveness alone will not benefit this stratum, so there should also be strategies aimed at specifically reducing poverty.

This perspective emerged as a reaction to the macroeconomic structural restructuring that developing countries underwent in the 1980s and 1990s. The distributional losses were too excessive to ignore, which necessitated the creation of new social security policies and more reasonable growth strategies to restore fiscal balance and economic efficiency (and secure repayment of debts to international financier.

In order to combat poverty, the central leadership has come up with a plan that develops innovative economic activities in underprivileged territories with a host of supporting policies (Zhou et al. 2014; Wang 2016; Gao 2014):

- Each poor village has been encouraged to develop its own special products, and special farming bases have been established to generate employment for the impoverished population;
- Local cooperatives of farmers and leading enterprises have received support so that they may act as the economy's driving force and strengthen their ties with poor households;
- Poor areas have received support in developing food processing and accelerating the integrated growth of agriculture, industry and the tertiary sector, so as to ensure that poor households will benefit more from the appreciation earnings of complete industrial and value chains;
- Marketing and promotion of farm production from underprivileged terrains is to be reinforced;
- The natural and cultural potential of poor areas will be explored to develop countryside tourism;
- Local hydropower, coal, oil, gas and other resources will be managed rationally and systematically, with simultaneous adjustments to the policy on distributing the returns from resource development;
- A hydropower benefit-sharing mechanism will be introduced, with the profits from power generation going, first, to relocation of residents from reservoir zones and, later, to improvements in these areas;
- State-owned enterprises and the private sector will be encouraged to establish investment funds aimed at industrial development in underprivileged areas, and, using the market, to attract more people to join in the development of local resources, industrial parks and new cities and towns in poor areas.

Photo voltage (PV) projects have been an effective tool in China's poverty alleviation efforts. Supported by dependable technologies, such clean power projects can generate stable income for the poor and contribute to local development. The essential equipment is easy to install in small solar stations operated by households or villages, or in bigger solar plants. They could supply power to agriculture and forestry as well. PV projects appear to be particularly suitable for locations with great amounts of sunshine. This also fits with China's strategy of promoting clean and low-carbon energy (Porteous 1991).

PV projects of varying scales can be altered to suit local conditions (Hasan et al. 2011):

(1) Firstly, 3–5 KW solar facilities for household use could be placed on rooftops or in the courtyards of rural houses, with the property rights and incomes belonging to the households;

(2) Additionally, 100–300 KW solar installations for village use could be erected on communally owned land, with the property rights belonging to the village and the benefits shared by the village as a whole, and poor households;

(3) PV plants could be based on greenhouses, with the property rights jointly owned by the investing enterprises and the poor households;

(4) Additionally, 10 MW-plus PV plants could extend on unexploited mountain slopes, with the property rights owned by the investing enterprises, which would donate some shares from which the benefits would be distributed to poor households by the local government.

In view of the environmental pollution caused by local waste products, it is important to determine how to effectively use the Internet to purchase waste products and incorporate recycling practices into the business models. Below are some ideas that top managers in different fields could introduce in their companies to achieve better sustainable results (Chao 2011; Xie 2016; Zhu 2008).

1. Explore the development path and mode of "Internet + Waste recycling". Considering the short development time of Internet recycling, it is necessary to evaluate existing practices and summarize the successful ones, analyze practical problems, distinguish the role of government and market, and guide the innovation of the recycling mode. Moreover, the profit models of enterprises must also be updated so as to ensure the sustainable development of the industry after the policy support is secured (Zhang 2017).

2. Expand the scope of operations of environmental protection enterprises. Build a new pattern for trading platforms managing transactions, payment platforms managing payments and financial institutions managing funds. Through market analysis, establish price index to guide product valuation; establish a credit system and form a credit evaluation system by means of industrial and commercial certification. Integrate information flow, logistics and capital flow, make an online layout and build industrial parks offline to form a new development model of renewable resources industry. For example, one company has built an industrial chain with Internet sanitation at the core; with the help of online apps and offline recycling boxes, another promotes the concept of environmental protection, instructs residents on how to sort and dispose of garbage at designated points and forms an "Internet + classification and recycling" model. Relying on the "Easy regeneration network", it builds an O2O platform for renewable resources, serving three types of businesses in the industrial chain: professional recyclers, rough processing and deep processing.

3. Encourage the use of the Internet, big data, and Internet of Things, information management public platforms and other tools of information collection, data analysis and flow monitoring, optimization of reverse logistics, network layouts, online recycling and offline logistics. Improve the information the public receives on recycling and general awareness of it. Automate recycling processes to change the traditional "small, scattered and poor" recycling model.

4. Strengthen capacity building. Improve the ability of intelligent identification, positioning, tracking, monitoring and management of waste recycling. Recycling enterprises should assume social responsibility, train or guide the quality of recyclers in various ways, improve their skills and create greater value through Internet recycling.

5. Extend intelligent recycling to Internet recycling. Before the "Internet Plus" campaign was launched, some Chinese websites began to explore online recycling, while some enterprises adopted similar "Internet Plus thinking".

6. Improve policy measures. Create preferential policies such as tax exemption for Internet waste recycling platforms, intensify the price reform, form a reasonable price comparison between traditional resources and renewable resources and improve the level of recycling and dismantling of scrapped cars, recycling and utilization of electronic waste and low-price waste (Callan and Thomas 2006; Zhang 2017).

"Internet + Waste recycling" has given rise to new forms of business. Convenience and low cost are the key to the sustainable operation of recycling platforms. Only through continuous exploration, necessary support and innovation of business models can economic, social and environmental benefits be organically unified.

## 4. Recommendations

The results of this research allow us to conclude that global social sustainability and inclusion, especially in the current turbulent and unpredictable conditions, require long-term management decisions made at the state level and targeted managerial solutions for the most vulnerable countries. It has been proven that Belgium, Czechia, the USA, China and France have the highest values of the social component of global inclusive circular economy. Therefore, they can and should make suggestions on how to overcome these

discordant social and environmental disbalances. Moreover, this paper offers individual indicators for each studied country as a supplement to the potential managerial decisions for the future. In particular, indicators that lower the value of the social component are the weaknesses of a country's economic policy. These are the "warnings" that even the socially developed countries should heed when shaping state strategies to combat social inequality. In addition, the spotlight is on the countries designated by the World Bank as assistance and aid program beneficiaries. Therefore, the article's novelty lies in that the identified countries and indicators can act as the basis for creating road maps for states supported by the World Bank. Thus, the paper offers individual blocks that can used to build a road map and mechanisms for the use of top managers and countries as a whole.

## 5. Limitations

This paper describes the social component of the global inclusive circular economy index. The authors have developed an entire methodology; however, this article focuses only on its social aspect. The aim of such choice is to identify the socially sustainable countries that can develop sustainable social policy. Thus, future research might focus on other areas, such as the economic, ecological, or circular components. Altogether, this would make it possible to evaluate the current situation and build comprehensive mechanisms for managing critical situations.

## 6. Conclusions

Thus, the results of this article make it is possible to build a road map of social inclusion for the most vulnerable countries. It should be based on a ranking of countries by the social component of the global inclusive economy, as the countries that are in the top (Belgium, the Czech Republic, the USA, China, France and others) can offer strategic directions for solving the problem of social inclusion in conditions of global imbalances. This issue will be especially relevant for countries such as Peru, Bangladesh, Congo, India, Nepal, Nigeria, Panama and the Philippines. After all, the World Bank has determined that these countries are vulnerable, and the issue of global social inclusion is acutely overdue in these areas.

The best way to solve the environmental problems associated with waste management is to reduce its volume. Therefore, it can be argued that waste processing contributes to the reduction of pressure on ecosystems both at the input and output stages. An alternative to recycling is incineration, which releases energy. However, this process is one-time. When the material is burned, it is lost forever. By recycling, you can design several methods of operation. The real value of recycling is determined within the framework of an integrated material, energy and waste-management system.

At the same time, other vital technologies—for example, the Internet of Things (IoT) or artificial intelligence—serve as the basis for the growth and enhancement of circular economy. The scale of risks, threats and, consequently, challenges in reducing waste volume and managing the need for this process confirms the presence of significant opportunities for business. The digitization requires a "new infrastructure" that would connect all devices to the Internet of Things. This involves deploying a network of sensors to collect data and provide analysis of the flow of materials, products and information. These data are then analyzed to justify decisions about resource consumption and the need to develop certain systems.

So, the fundamental concept of an inclusive circular economy centers on frugal use of material resources and sensible management of labor force. In the conditions of an all-encompassing circular economy, development is driven by human capital rather than excessive and ever-growing extraction of natural resources.

In its simplest definition, such an economy is "low-carbon", efficient and "clean", and is based on consumption and results obtained through exchange, circularity, cooperation, solidarity, sustainability, the use of opportunities, etc. In this context, diversified sustainable development provides for the expansion of options for national economies, application of

targeted and appropriate fiscal and social policies, support for institutions whose activities are aimed at protecting social and ecological values.

**Author Contributions:** Conceptualization, A.K.; Resources, O.B.; Writing—original draft, I.Z.; Visualization, R.Z. All authors have read and agreed to the published version of the manuscript.

**Funding:** This research received no external funding.

**Data Availability Statement:** https://stats.oecd.org/, https://ec.europa.eu/eurostat, accessed on 18 November 2022.

**Conflicts of Interest:** The authors declare no conflict of interest.

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
