# Peer review of "Global Social Sustainability and Inclusion: The “Voice” of Social and Environmental Imbalances"

_jrfm, doi:10.3390/jrfm15120599_

Round 1
Reviewer 1 Report
I did not necessarily find this paper to be significantly novel outside of the authors rankings of countries, which I believe warrants its publication.
Author Response
Good morning,
Thank you very much for your review of the article and positive feedback.
The English language of the article was corrected by an English reviewer (C2 language level).
With best wishes, Doctor of Economics, Professor Iryna Zvarych.
Reviewer 2 Report
The manuscript title “Global Social Sustainability and Inclusion: the "voice" of social and environmental imbalances” The article is in my domain and it’s pleasure to give the feedback on your manuscript. To me it’s well written and well structured. However, few observations are mentioned below to fix them for the next step.
· Try to avoid acronyms from the abstract and refinement the abstract.
· Reflection the Debate Literature review (LR), should be mix up with the old and new references. Nevertheless, there is a still strong need have enhance through to add LR from years 2021, 2022. Few suggestions are below but not limited to add strongly recommended to add latest references.
1. https://doi.org/10.1007/s11356-022-23238-8
· Methodology section seems fine.
· Strengthen Discussion section to add more paragraph.
· What is novelty of this study?
· Limitations and future scope?
· During review, I noticed that some references are not cited in the text and also some papers cited in the text but author(s) probably forgot to provide their references. Please recheck your entire references and citations.
· Further, author(s) should check the grammatical and English errors. I suggest author(s) to proof editing to the entire manuscript, it will significantly help to improve English language.
I hope these comments are useful to improve the quality of the paper.
Good luck and all the very best!
Author Response
Good morning,
Thank you very much for your review of the article and positive feedback.
- According to the “there is a still strong need have enhance through to add LR from years 2021, 2022”
I added this new sources in text and references;
- Strengthen Discussion section to add more paragraph.
Discussion section is Strengthed and added more paragraph.
- What is novelty of this study? Limitations and future scope?
(this paragraph is added)
Introduction:
The problem of global social inclusion, inclusive economy arose during the study of the problem of recycling and the circular economy in general. After all, people are part of nature (Daly, 2009) and economic problems are inextricably linked with ecology, ethics and social approach. The problem of harmonizing life on earth in harmony, peace, complete, so to speak, in the modern way of inclusiveness is described (Commoner, 1992). But most of the reviewed works make us think about global permanent inclusion, the role of different countries in this. What factors in one way or another affect the isolation of the social component in global sustainable inclusiveness? Which countries have the most pronounced social component? Is there a rating of such countries. The authors thought about such questions and offered to give answers in this article.
The novelty of this article will be the highlighting of the ranking of countries according to the social component, based on the author's methodology, and the emphasis on social inclusiveness in them.
- During review, I noticed that some references are not cited in the text and also some papers cited in the text but author(s) probably forgot to provide their references. Please recheck your entire references and citations.
All references have checked and added in text and some numbers in list as well.
- Further, author(s) should check the grammatical and English errors
The English language of the article was corrected by an English reviewer (C2 language level).
With best wishes,
Doctor of Economics, Professor Iryna Zvarych.

Reviewer 3 Report
- Similarity index is quite high (total 44%). Please check the attached file.
- Many sentences in the abstract section make little sense and are not following an abstract structure. There are many sentences which are more appropriate in an introduction or literature review. Especially the last few sentences.
- Figure 1 is constructed by authors. But the authors did not explain how they came up with the concept. There is no literature review properly cited to justify the thinking process of developing the concept.
- The results section does not have any introduction of what is coming. Readers should be guided by some explanation of the results logics and structure before jumping to explain what the tables and figures are showing.
- There is a typo in figure 4. caption.
- The source of external figure should be cited by the source author/institution. not by the URL link of the website.
I'm sorry I cannot recommend this paper for publication as it has major flaws and full of technical errors. This manuscript was not crafted carefully, have low originality, and lacking of structure and content.

Author Response
Good afternoon,
Fits of all thank you so much for your detailed explanation.
I tried to take everything into account.
So,
- First of all, I assigned the article to an English reviewer (C2 language level), worked first of all to remove plagiarism according to the report, so everything was changed through the text.
- Second, all sources from the bibliography were used.
- Thirdly, I added the necessary paragraphs to the introduction, results and conclusions to the article to explain table an results.:
Introduction:
The problem of global social inclusion, inclusive economy arose during the study of the problem of recycling and the circular economy in general. After all, people are part of nature (Daly, 2009) and economic problems are inextricably linked with ecology, ethics and social approach. The problem of harmonizing life on earth in harmony, peace, complete, so to speak, in the modern way of inclusiveness is described (Commoner, 1992). But most of the reviewed works make us think about global permanent inclusion, the role of different countries in this. What factors in one way or another affect the isolation of the social component in global sustainable inclusiveness? Which countries have the most pronounced social component? Is there a rating of such countries. The authors thought about such questions and offered to give answers in this article.
The novelty of this article will be the highlighting of the ranking of countries according to the social component, based on the author's methodology, and the emphasis on social inclusiveness in them.
Results
To fulfill the task, the indicators of the relevant countries and the statistics of the Organization for Economic Cooperation and Development for 28 member countries of the organization from 1995 to 2020 were used: Australia, Austria, Belgium, Great Britain, Greece, Denmark, Estonia, Israel, Canada, China, Latvia, Lithuania, Luxembourg, Mexico, Netherlands, Germany, New Zealand, South Africa, South Korea, Poland, USA, Turkey, Hungary, Ukraine, Finland, France, Czech Republic, Japan. The analysis was carried out on the basis of the GNU Regression, Econometrics and Time-series Library (Library for regressions, econometrics and time series) - an applied software package for econometric modeling, part of the GNU project.
Conclusions
Thus, it can be said that based on the results of the article, in the following studies it is possible to build a road map of social inclusion for the most vulnerable countries. The basis for such a study will be the ranking of countries by the social component of the global inclusive economy. It is the countries that are in the top (Belgium, the Czech Republic, the USA, China, France and others) that can offer strategic directions for solving the problem of social inclusion in conditions of global imbalances. This issue will be relevant for such countries as Peru, Bangladesh, Congo, India, Nepal, Nigeria, Panama, and the Philippines. After all, it is precisely these countries that the World Bank has identified as vulnerable and those in which the issue of global social inclusion is acutely overdue.
- In the material that is in the table 4 and 5, I added a source from the reference, since the text is very accurate , we want to admit this neadable information from the sourse WORLD BANK, WE PUT 28(#IN REFERENCES), more over another references were added.
With best wishes, Doctor of Economics, Professor Iryna Zvarych.

Round 2
Reviewer 3 Report
Although the manuscript has been edited by the authors to some extents, it is obvious that the English has not been sent to a professional editor. Moreover, minor corrections such as reving the URL links as sources from the text has not been done either.
Author Response
I sincerely thank you for the clarification and comments.
So, the whole article has passed the English language check, it has been completely corrected by a different reviewer than the previous time. I hope you can see it.
In addition, references are removed from the text.
- ABSTRACT. The last sections of the abstract is still quite generic. It doesn’t not reflect the changes made in the text in the managerial implications section. It should be revised accordingly.
Abstract:
Background: Global environmental and social research strengthens the protection of people and the environment, develops national capacity for social and environmental management and en-ables significant progress in terms of transparency, accountability, non-discrimination and public participation. The support of the general public plays a key role as it contributes to making public institutions more transparent, accountable and efficient and promotes ground-breaking solutions to complex development challenges. Citizen engagement seems particularly vital throughout the crises such as the COVID-19 pandemic, as the efficiency of response efforts may frequently depend upon micro-level behavioral changes. Objective: To provide a complex evalu-ation and rating of countries based on the social component of the global inclusive circular economy, taking into account the shocks and reverberations experienced by the economy as a whole caused by the Covid-19 and war in Ukraine. The results of paper are presented as a global ranking of countries based on the social component of the global inclusive circular economy. They confirm the high value of this component in the integrated indicator, which validates the hy-pothesis that inclusiveness is a necessary aspect of the global circular economy. The research results identify the countries capable of offering the best management solutions to social dis-balances and other weaknesses, as well as the countries in need of model examples to tackle these issues.2. INTRODUCTION. The same problem can be found in the introduction. There are some clear sentences that highlight the theoretical implications of the study, however, very few sentences show that this study also is able to provide insights for managers and practitioners.
- LITERATURE REVIEW, METHODOLOGY, DISCUSSIONS AND THEORETICAL CONTRIBUTIONS have been strongly revised thank to the very useful comments of reviewers and the editor.
- MANAGERIAL IMPLICATIONS. There is still the need to strengthen the managerial implications.
While you mentioned some important managerial contributions, some are quite generic.
Instead of focussin on generic points, you should try to answer to the following questions: What kinds of objective evidence can you offer that would make industry leaders sit up and pay attention to your study? What makes this topic a big deal right now, and perhaps in the immediate future?
For the first time, this paper presents objective evidence in the form of a rating of countries based on the social component and loosely correlated derived indicators. Thus, the latter do not make a significant impact on solving disbalances in the studied countries. Such analysis is conducted for the first time and is therefore useful to top managers and higher-up officials shaping state policy.
This topic is important now and will undoubtedly be important in the near future in light of the ongoing Covid-19 pandemic, the war in Ukraine, and mass migration of Ukrainians to other countries, which all exert social pressure on the local populace, exacerbate gender inequality and violence … This topic is an inexhaustible well of questions and answers. Thus, we must set careful limits to our research, as we have chosen to focus on the social component, i.e., social inclusiveness, only.
Recommendations:
The results of this research allow us to conclude that global social sustainability and inclusion, especially in the current turbulent and unpredictable conditions, require long-term management decisions made at the state level and targeted managerial solutions for most vulnerable countries. It has been proven that Belgium, Czechia, the USA, China and France have the highest values of the social component of global inclusive circular economy. Therefore, they can and should make suggestions on how to overcome these discordant social and environmental disbalances. Moreover, the paper offers individual indicators for each studied country as a supplement to the potential managerial decisions for the future. In particular, indicators that lower the value of the social component are the weaknesses of a country’s economic policy. These are the “warnings” that even the socially developed countries should heed when shaping state strategies to combat social inequality. In addition, the spotlight is on the countries designated by the World Bank as assistance and aid program beneficiaries. Therefore, the article’s novelty lies in that the identified countries and indicators can act as the basis for creating road maps for states supported by the World Bank. Thus, the paper offers individual blocks that can used to build a road map and mechanisms for the use of top managers and countries as a whole.
- LIMITATIONS. This section does not enrich the paper. It repeats the discussion and does not reveal how analysing the paper’s limits can help other authors to add value to the field of the research.
I suggest that “limitations” should be enriched with a future research agenda.
Limitations
This paper describes the social component of the global inclusive circular economy index. The authors have developed an entire methodology; however, this article focuses only on its social aspect. The aim of such choice is to identify the socially sustainable countries that can develop sustainable social policy. Thus, future research might focus on other areas, be it the economic, ecological, or circular components. Altogether, this would make it possible to evaluate the current situation and build comprehensive mechanisms for managing critical situations.
More over the reccomendations and limmitations are added in text.
Thank you again
Sincerely

Round 3
Reviewer 3 Report
I have nothing else to add.